# Injury Severity and Psychological Distress Sustained in the Aftermath of the Attacks of 11 September 2001 Predict Somatic Symptoms in World Trade Center Health Registry Enrollees Sixteen Years Later

**DOI:** 10.3390/ijerph17124232

**Published:** 2020-06-13

**Authors:** Howard E. Alper, Lisa M. Gargano, James E. Cone, Robert M. Brackbill

**Affiliations:** New York City Department of Health and Mental Hygiene, 30-30 47th Avenue, Room 414, Long Island City, NY 11101, USA; lisa.gargano@gmail.com (L.M.G.); jcone@health.nyc.gov (J.E.C.); rbrackbi@health.nyc.gov (R.M.B.)

**Keywords:** World Trade Center, 9/11, disaster, injury, mental health, somatic symptoms

## Abstract

The World Trade Center attacks of 11 September 2001 (9/11) have been associated with the subsequent development of chronic diseases. Few studies have investigated the burden of somatic symptoms on attack victims, or the association of such symptoms with exposure to the 9/11 attacks. World Trade Center Health Registry (Registry) enrollees who were present south of Chambers Street during or immediately after the 9/11 attacks and who provided consistent answers regarding injury sustained on 9/11 were followed prospectively for up to 16 years post-9/11/01. We employed linear regression to evaluate the associations between injury severity, psychological distress and somatic symptoms in 2322 persons who completed all four Registry surveys and a subsequent Health and Quality of Life survey. Twenty-one percent of subjects had a “very high” burden of somatic symptoms, greater than in populations not exposed to a disaster. Somatic symptoms exhibited a dose-response association separately with injury severity and psychological distress trajectories. Victims of the 9/11 attacks suffer from a substantial burden of somatic symptoms which are associated with physical and psychological consequences of exposure to the attacks. Physical and mental health professionals need to work together when treating those exposed to complex disasters such as 9/11.

## 1. Introduction

The terrorist attacks on the World Trade Center (WTC) on 11 September 2001 resulted in over 2800 deaths, and many thousands more injured. The dust/debris cloud generated by the collapse of the WTC towers enveloped many survivors and responders, who also may have been injured or witnessed horrific events such as seeing airplanes strike the towers. These exposures led to the development of a variety of physical and mental conditions, such as asthma [1,2,3,4,5], post-traumatic stress disorder (PTSD) [6,7,8,9], heart disease [6,10], stroke [8], and cancer [11,12].

Literature has shown that, among the general population with experience of serious injuries, anxiety, depression, or posttraumatic stress disorder (PTSD) can persist for years after the injury [13]. Being injured during the 9/11 attacks significantly increased the likelihood of PTSD years later [3]. Brackbill et al. [14] found that the likelihood of being diagnosed with respiratory or circulatory diseases 10–11 years after the disaster exhibited a dose-response relation with the number of types of injuries they sustained on 9/11. A recent study showed that heart disease (e.g., heart attack, angina) in particular exhibited a strong dose-response relationship with the number of injuries sustained on 9/11 10–11 years after the disaster [6].

A recent qualitative study by the WTCHR of persons who sustained an injury on 9/11 reported various decrements in quality of life [15]. Subjects reported pain, gastrointestinal problems, functional limitations and disabilities, and limitations on social activities. This qualitative study informed a subsequent investigation of physical and mental functioning in Registry enrollees injured on 9/11, which demonstrated that poor physical functioning 16 years after the 11 September 2001 attacks, as measured by the SF-12 scale, exhibited a dose-response relationship with increasing severity of injury on 9/11 [16].

Somatic symptoms constitute the experience of various bodily symptoms whose causes can be difficult to determine. Severe somatic symptoms are described as somatoform disorders and consist of symptoms such as loss of vision or hearing. Less severe, but more prevalent, are symptoms such as gastrointestinal complaints, bodily pain (e.g., arms back, headache), cardiopulmonary effects (e.g., chest pain, dizziness), and fatigue (e.g., sleeping troubles, low energy). Somatic symptoms accompany many physical [17,18] and mental disorders [19] and contribute substantially to the cost of health care [20].

Somatic symptoms are a class of outcomes that have not been evaluated in reference to 9/11 but have been the subject of studies for other disasters. For example, a study of 162 persons exposed to the Great Midwestern Flood of 1993 found that 25% of subjects experienced new somatic symptoms after the flood [21]. A review of 57 studies of somatic symptoms after disasters worldwide from 1983 to 2003, reported that prevalence of Medically Unexplained Physical Symptoms (MUPS) varied by which were dominant symptoms and latency after the disaster [22]. For example, symptom prevalence three months post disaster ranged from 10% to 65%, while some symptoms (fatigue, poor digestion, trouble breathing, skin problems, or cough) persisted at least six years post disaster with prevalence ranging from 10% to 50%. Predictors of MUPS include disaster-related consequences such as injury, and post-disaster variables such as coping style. The latter may be affected by mental health conditions that develop post-disaster, such as PTSD or psychological distress. While research has investigated the association of PTSD with somatic symptoms [23], no studies to our knowledge have looked at the relation between psychological distress and somatic symptoms. Furthermore, there has been little research on the association of injury number, type, or severity with somatic symptoms [24]. Moreover, we know of no research exploring the association between both psychological distress and injury with somatic symptoms. Finally, the association between psychological distress and injury with somatic symptoms in those exposed to the 9/11 attacks has not been investigated.

The goals of this study were to: (1) evaluate the prevalence of somatic symptoms among those exposed to the 11 September 2001 attacks, with comparison to other groups not exposed to a disaster, (2) estimate the association between the severity of injuries sustained on 9/11, and (3) history of non-specific psychological distress, with somatic symptoms after controlling for pre-9/11 history of physical and mental health diseases, demographics, and other risk factors.

## 2. Materials and Methods

### 2.1. Data Source and Study Design

The World Trade Center Health Registry (The Registry) was created in 2002 to monitor the physical and mental health consequences of those exposed to the terrorist attacks of 11 September 2001. Enrollees included rescue/recovery workers, residents, area workers, passersby, and students/staff of local schools. The initial Registry survey was conducted in 2003–2004 (wave 1), and subsequently in 2006–2007 (wave 2), 2011–2012 (wave 3), and 2015–2016 (wave 4). The methods used by the Registry have been described in detail in previous publications [3,25]. The Registry was approved by the institutional review boards of the Centers for Disease Control and Prevention and the New York City Department of Health and Mental Hygiene.

The data source for this study was derived from a Registry sub-study, “The Health and Quality of Life 15 Years After 9/11 Survey” (HQoL survey) [26]. The HQoL study was approved by the institutional review board of the New York City Department of Health and Mental Hygiene. The HQoL study was completed after the wave 4 survey. Inclusion criteria for this study included: (a) completing all four WTCHR survey waves, (b) being at least 18 years of age on 9/11, and (c) speaking English. This sample consisted of two groups. The injured group included persons who reported on Wave 1 that they sustained one or more of five types of injuries on 9/11: cut, abrasion or puncture wound; sprain or strain; burn; broken bone or dislocation; and concussion or head injury. Those who reported “other injury” or an “eye injury” only were not included. The second group consisted of a non-injured comparison group of randomly selected persons who did not report any type of injury including “other” or “eye injury”.

### 2.2. Analytic Sample

The sample for this study was derived from the HQoL study data source by applying the following additional inclusion/exclusion criteria: (1) the enrollee was south of Chambers Street on 9/11, (2) the enrollee reported being injured on both the wave 1 and HQoL surveys, and (3) the enrollee had non-missing values of the Kessler 6 psychological distress sum score (see below) at waves 1, 2, and 4.

### 2.3. Study Variables

#### 2.3.1. Outcome Measure

The severity of somatic symptoms was assessed using the Somatic Symptom Scale-8 scale (SSS-8) [27]. The SSS-8 consists of eight questions, each measured on a Likert scale from 0 (“not at all”) to 4 (“very much”). The item scores are summed to create the scale, which can range in value from 0 to 32. The scale has excellent psychometric properties and good reliability (Cronbach’s a = 0.81) [27]. The SSS-8 scale possesses good construct validity, as it correlates with depression (r = 0.57), anxiety (r = 0.55), and general health status (r = −0.24) [27]. Somatic symptoms severity categories were taken from Gierk et al.: no to minimal (0–3), low (4–7), medium (8–11), high (12–15), and very high (16–32) somatic symptom burden [27]. The factor structure obtained from the SSS-8 scale reflects gastrointestinal (stomach or bowel), pain (arms, legs, joints, or headache), fatigue (tiredness, trouble sleeping), and cardiopulmonary (chest pain, shortness of breath, dizziness) components of general somatic symptom burden [27]. We chose the SSS-8 scale because it is well-validated and is briefer than scales such as the PHQ-15 [28,29] 2.3.2. Exposure Variables

Level of injury severity. This was categorized as none, low, medium, or high. None was defined as having reported no injury at 9/11 on the HQoL survey. Low injury severity was defined as having reported being injured at 9/11 on the HQoL survey, but answering no to all the following injury severity questions: (a) Spent the day in a bed, chair, or couch any time during the week after your injury, (b) Used a cane or crutch to help you walk any time during the week after your injury, (c) Spent time in a wheelchair any time during the week after your injury, (d) Where did you receive treatment for the most serious of your injuries (hospital/ER, doctor’s office, other, not applicable), (e) Have you ever received physical therapy, and (f) Did you have surgery for your injury. Medium injury severity was defined as having reported being injured at 9/11 on the HQoL survey, answering yes to all questions a–c above and no to all questions d–f. High injury severity was defined as having reported being injured at 9/11 on the HQoL survey and answering yes to any of the questions d–f) above. This is summarized in Brackbill et al. [16].

Mental health status. This was represented for the post-9/11 period by the trajectories (see below) of non-specific psychological distress, as measured by the Kessler (K6) instrument [30], a shortened form of the Kessler K10 scale. The K6 asks respondents to report how often they experienced six emotional states related to psychological distress in the past 4 weeks. The responses range from “All of the time”, “Most of the time”, “Some of the time”, “A little of the time”, and “None of the time”. The total score was obtained by the summation of each of the six emotional states related to psychological distress where “All of the time” = 4, and “None of the time” = 0. Total scores ranged from 0 to 24, where 0 indicated the lowest degree of psychological distress, and 24 indicated the highest. Individuals with a score greater than 13 were considered to have severe psychological distress (Kessler et al., 2002, [30]). The K6 has high internal consistency and reliability (Cronbach’s alpha = 0.89), and has minimal bias with regards to demographic features such as sex and education (Kessler et al., 2002, [30]). The K6 was administered at Waves 1, 2, and 4. We could not use the K6 data from Wave 3 because of an inadvertent measurement error, but the data from Waves 1, 2, and 4 were enough to form K6 trajectories.

The K6 trajectories cover the period wave 1 to wave 4. Injury severity includes events that occurred in the first weeks after the 9/11 attacks, and other events (e.g., surgery, hospitalization for other reasons) that could have occurred up to wave 4.

#### 2.3.2. Covariates

Mental health status pre-9/11 was treated as a dichotomous variable, such that if the enrollee self-reported PTSD, anxiety, or depression the mental health variable was set to 1, whereas if the enrollee reported none of these conditions, it was set to 0.

Physical health status. This was represented by a single variable, covering both pre-9/11 and post-9/11 periods, corresponding to the number of chronic diseases self-reported by the enrollee, including high cholesterol, hypertension, gastro-esophageal reflux disease (GERD), asthma, or sleep apnea. The physical health variable consisted of the count of positive responses to having any of the above diseases either before or after the 9/11 attacks. This was categorized as 0, 1, or 2 or more diseases.

Covariates included the following sociodemographic characteristics: gender (male, female), age at 9/11, race (white non-Hispanic, black non-Hispanic, Hispanic, Asian non-Hispanic, other), education (less than high school, high school/GED, some college/college grad/post-grad), income (0 ≤ USD 25 K, USD 25 ≤ USD 50 K, USD 50 ≤ USD 75 K, USD 75 ≤ USD 150 K, ≥ USD 150 K), eligibility group (rescue/recovery, resident, area worker, passerby, student/staff), and marital status (married/living together vs. divorced/separated/widowed).

Risk or protective factors were social support at wave 4 and stressful events before wave 3. Low social support is associated with increased severity of mental conditions such as PTSD and depression, and potentially with somatic symptom severity [31]. Social support was defined by the sum of the answers to five questions, each of which can range from 0 (“none of the time”) to 4 (“all of the time”): “How often is someone available to take you to the doctor if you need to go?”, “To have a good time with you?”, “To hug you?”, “To prepare your meals if you are unable to do it yourself?”, and “To understand your problems?”. The total score can range from 0 to 20, and low social support was defined as a total score below 15.

Stressful events can exacerbate mental health conditions that may contribute to somatic symptoms [32]. Having stressful events was based on a sum of endorsement of up to six events (such as unable to pay bills, lost a job, family problems), expressed categorically as none (0 events), low/medium (1 or 2 events), and high (greater than 2 events) [33].

### 2.4. Statistical Analysis

The frequency distributions for the exposure variables (injury severity, psychological distress trajectories), the sociodemographic variables, and risk factors were calculated. The mean SSS-8 score and its standard deviation were also computed for each category of the above variables.

The frequency distribution over the five somatic symptoms categories (none/minimal, low, medium, high, very high) derived from the SSS-8, was obtained for the following: (1) the full analytic sample of the present study, and (2) a nationally representative German sample [27].

The trajectories for psychological distress were obtained using the free program PROC TRAJ. This groups enrollees with similar patterns of K6 over time into mutually exclusive trajectories, with input from the investigator. In particular, the user can choose the number of trajectory groups, and the mathematical form of the trajectory (linear, quadratic, cubic). By varying these parameters and using both statistical (e.g., BIC or Bayesian Information Criterion) and subject matter criteria to evaluate a series of potential models, the user can arrive at a final model. This process is described in detail in a previous Registry publication [34].

We conducted unadjusted and adjusted trajectory analyses of psychological distress over time. The unadjusted models only included the K6 scores, whereas the adjusted models incorporated time-varying and time-invariant covariates. Trajectory analysis enabled (1) prediction of membership in trajectory groups, (2) prediction of trajectory shapes, and (3) direct comparison of two trajectories. Time invariant covariates predict group membership, while time variant covariates predict trajectory shape.

The user defines the number of trajectories. Additionally, the user selects the order (linear, quadratic, cubic) of the regression equation for each trajectory group. The optimal number of groups and equation order are determined by incremental decreases in Bayesian information criteria (BIC) from the test model to the previous model. PROC TRAJ can accommodate a variety of distributions; the present study selected a censored normal distribution as K6 is a psychometric scale. For each enrollee, PROC TRAJ outputs the probability for likelihood of assignment to each group; each enrollee is assigned to the group with the highest probability. PROC TRAJ operates on complete-case analysis for covariates: enrollees missing any time variant or time invariant covariates were excluded from the analysis.

Model selection was conducted iteratively, starting with the number of groups set to one, and the order set to quadratic. The number of groups was increased in each iteration of the model; the BIC was monitored for substantial changes. We did not consider model orders greater than quadratic since we only had three data points for the K6. Once the optimal model was selected, posterior probabilities and group size estimates were obtained from the PROC TRAJ output. Next, the time varying and time invariant covariates were included in the above optimal model. This final model produced the K6 trajectories employed in all analyses. PROC TRAJ model selection was also informed by previous trajectory analyses on other psychological variables, particularly PTSD. For this, the following trajectory types are often observed: resilient, low stable, moderate increasing, high decreasing, and chronic. We hypothesized similar results for psychological distress. After using PROC TRAJ on the K6 data as described above, we observed the first four of the above five trajectory types: resilient, low stable, moderate increasing, and high decreasing.

Linear regressions were performed to determine the associations of the continuous SSS-8 scale score with injury severity, psychological distress trajectories, sociodemographic variables, and risk factors. Regression coefficients and their 95% confidence intervals were taken as the measure of association and its statistical significance. We examined the injury severity-psychological distress trajectory interaction by including their product term in a version of the main regression. If the associated *p*-values < 0.05 an interaction was assumed to exist. We also tested for linear dose-response relationships of SSS-8 scores with injury severity and psychological distress trajectories by treating these exposures as continuous (not categorical) and calculating the *p*-value of the regression coefficient. If this was <0.05, we concluded a linear dose-response relationship. All calculations were performed using SAS version 9.4 (SAS Institute Inc., Cary, NC, USA).

## 3. Results

The mean SSS-8 scores are shown in Table 1. Somatic symptoms increase substantially with both injury severity (none: 7.7, low: 11.0, medium: 13.7, high: 14.3) and psychological distress trajectory (resilient: 5.0, low stable: 9.8, moderate increasing: 16.2, high decreasing: 15.2). Somatic symptoms decline with education (less than high school: 12.4, high school: 11.6, college: 8.5), as it does with income. Somatic symptoms scores are higher among those having more stressful events (three or more: 14.6, one or two: 11.5, none: 8.5).

The characteristics of the study sample (*n* = 2322) are also shown in Table 1. No, low, medium, or high injury severity were experienced by 67%, 4%, 16%, and 13% of study enrollees, respectively. The PROC TRAJ procedure identified four K6 trajectories: Resilient, low-stable, moderate increasing, and high decreasing. The resilient group was characterized by mean K6 scores below two through waves 1–4. The low-stable exhibited average K6 scores between 3–5 over waves 1–4. The moderate-increasing trajectory had an initial mean K6 value of 8 but increased to 13 by wave 4. Finally, the high-decreasing group had a wave 1 K6 mean value of 11, which decreased to below 7 by wave 4. These trajectory groups characterized 24%, 58%, 10%, and 8% of the study sample, respectively. There were 57% male, 51% aged 25–44 at 9/11, 77% white non-Hispanic, 66% college-educated, 58% earned above USD 75 K, 58% area workers, and 69% were married/living together. A total of 56% reported low social support. The overwhelming majority (95%) of enrollees did not report stressful events, and 62% reported at least one chronic disease pre- or post-9/11.

The somatic symptoms category distributions are presented in Figure 1 for the following: (1) present study, full analytic sample (orange) and (2) nationally representative German sample (blue) [27]. Several features are notable. First, 21% of the present study sample were in the “very high” somatic symptom severity category, while only 47% were in the “none” or “low” categories. For the nationally representative German sample, only 2–3% were in the “very high” category, while 88% were in the “none” or “low” categories.

Results for the linear regressions are presented in Table 2. Somatic symptoms exhibited a modest dose-response relationship with injury severity (for low injury severity β = 1.5, 95% confidence interval (CI) 0.3–2.7; for medium injury severity β = 3.1, 95% CI 2.4–3.8; for high injury severity β = 3.7, 95% CI 2.9–4.5; dose-response test *p* < 0.001) and a strong dose-response relationship with the psychological distress trajectories (Low stable β = 3.3, 95% CI 2.7–3.9; moderate increasing β = 8.2, 95% CI 7.2–9.2; high decreasing β = 6.4, 95% CI 5.4–7.4; dose-response test *p* < 0.001). The interaction between injury severity and psychological distress trajectories was not significant (*p* = 0.39).

## 4. Discussion

To our knowledge, the present study is one of the first to investigate the prevalence and predictors of somatic symptoms among those present at the attacks of 11 September 2001. We found a prevalence of “very high” somatic symptom levels of 21%, significantly higher than in a nationally representative German sample (2–3%). Though the SSS-8 scale has not, to our knowledge, been applied to a representative sample of the United States population, it is likely the prevalence of “very high” somatic symptom levels would approximate the above German sample.

We showed that somatic symptoms exhibited a substantial and statistically significant dose-response relationship with both the severity of injury sustained on 9/11 and trajectories of psychological distress, two factors that encompass a 15-year period. This is consistent with the hypothesis that both physical and mental health conditions contribute over time to the development of somatic symptoms.

The mechanism leading from injury to somatic symptoms is unclear. Nader [35] proposed that traumatic injury can lead to peripheral inflammation, which can produce or exacerbate somatic symptoms. Irwin [36] summarized evidence that activation of the pro-inflammatory cytokine network in response to tissue injury can underlie the pathophysiology of somatic symptoms. These findings are also consistent with an effect of injury on somatic symptoms. Another theory [37] proposes that somatic symptoms develop as a response to mental health conditions that are not recognized by the person or remain untreated for other reasons. This is consistent with the finding of the present study of an independent (from the effect of injury severity) association between K6 psychological distress trajectories and somatic symptoms. Additionally, recent research [38] has shown that anxiety sensitivity might explain the co-morbidity between PTSD and somatic symptoms, consistent with both direct and mediated effects of injury.

The present study extends previous Registry research into the effects of the 9/11 attacks by investigating somatic symptoms. Specifically, while previous Registry studies showed an association between injury, PTSD, and heart disease [6,14] out to 10–11 years post-disaster, the present study demonstrated that injury severity and psychological distress trajectories are also associated with somatic symptoms measured 16 years after the 9/11 attacks.

The present study contributes to the literature on the prevalence and predictors of somatic symptoms. While several studies have investigated the association of PTSD with somatic symptoms [23], few if any have looked at the relation between psychological distress, arguably a more accurate description of one’s general psychological state, and somatic symptoms. Nevertheless, our study agrees with previous research in that increased severity of mental health disease was associated with increased severity of somatic symptoms. Furthermore, the association of injury characteristic with somatic symptoms has been little studied [24], but the present study agrees with previous research in that increased number or severity of injury was associated with increased severity of somatic symptoms.

The importance of somatic symptoms extends beyond the suffering of individuals. Somatic symptoms also contribute substantially to health care costs. They are estimated to account for 10–20% of total annual medical costs in the US [20], and for 10% of health care costs among working age people in the UK [39]. In the US, patients with somatic symptoms have twice as many primary care visits, specialty visits, emergency department visits, and hospital visits compared to patients without somatic symptoms [20]. The same study found that health costs for patients with somatic symptoms were also 2–3 times that of patients without such symptoms. The individual and societal costs associated with somatic symptom remain an important public health concern.

A major strength of the present study is the use of a prospective cohort, which allows us to investigate the association of exposures to, and sequelae of exposure to, the 9/11 attacks with later emerging conditions. Further, the sample was composed of injured and non-injured enrollees, which allowed determination of the association between injury severity and somatic symptoms. The present investigation also used well-validated measures of psychological distress, the K6 scale, and of somatic symptoms, the SSS-8 scale [27]. Finally, the longitudinal nature of the Registry cohort made it possible to measure both K6 trajectories and injury severity over a 15-year period.

A limitation of our study is that the list of injuries reported in Wave 1 might seem mild to moderate only. However, in the HQoL survey injured enrollees reported the circumstances of their injury [26]. In total, 9.8% of reported being hit by a falling object, and 8.3% reported hitting their head on an object. Further, in their answers to an open-ended question on how injury occurred enrollees provided details such as “a building fell on me” and “consumed in fire”. Since such injuries were likely to be more than moderate, we believe that this study includes enrollees with mild, moderate, and severe injuries.

A second limitation is that this study included only enrollees whose spoken language was English. While 95% of Wave 1 enrollees spoke English, excluding non-English speakers introduced a bias in our ability to generalize our results to the entire affected population.

Another limitation of our study was our inability to compare our somatic symptoms results to subjects representative of either New York City/State or the US, because the SSS-8 has not, to our knowledge, been used in such a survey. However, a recent study using data from the National Latino and Asian American Study (NLAAS) [40], a nationally representative survey of White, Latino, and Asian Americans adults, found that subjects on average endorsed only 0.5–1 somatic symptoms of fourteen possible symptoms. The NLAAS results could be consistent with those for the nationally representative German sample [27] described above, but there may be differences in ethnic and cultural endorsement of specific somatic symptoms, so caution in the above comparison is warranted.

Another limitation of our study was that data for exposures, outcomes, and covariates were self-reported. However, there has been good agreement between Registry findings based on self-report and those based on hospitalization data [11].

A further limitation is that the injury severity exposure variable was constructed using data from the HQoL survey, sixteen years after 9/11. It is possible enrollees provided inaccurate answers to the injury severity questions, such as receiving treatment or having surgery. However, we excluded all enrollees who provided inconsistent answers, between the wave 1 and HQoL surveys concerning being injured on 9/11.

An additional limitation was that the pre-9/11 mental health status variable encompassed only self-reported diagnoses of depression, anxiety, and PTSD. The definition excluded externalizing conditions such as substance use disorders, ADHD, and conduct and antisocial personality disorders. These conditions, encompassing impulsivity, aggression, and rule breaking manifestations, are associated with risks of exposure to trauma. Externalizing pathology, particularly of the antisocial variety, is associated with both diagnosed chronic disease and medically unexplained symptoms.

Another limitation is that our analytic sample included enrollees (70%) with at least one of the five most prevalent chronic diseases at wave 4. These could affect the associations between somatization and injury severity and/or psychological distress trajectories. We tested this hypothesis by creating a sub-sample of ~700 enrollees with none of the five chronic diseases. The associations between somatization and the two above exposures for this sub-sample were similar in size to associations obtained using the complete analytic sample.

Finally, the attrition observed in the Registry cohort could bias the associations we observed, though a recent study [41] found such bias to be small.

## 5. Conclusions

The present study shows that exposure to the events of 11 September 2001 was associated with experiencing somatic symptoms 16 years after the attacks. Specifically, the severity of injury and the psychological distress trajectories were associated with the degree of subsequent experience of somatic symptoms. Given the large fraction of enrollees who experienced “very high” levels of somatic symptoms, the present study implies that a substantial number of people present for the 9/11 attacks could currently be experiencing somatic symptoms, adding to an already significant burden of disease. Physical and mental health professionals need to work together when treating those exposed to complex disasters such as 9/11. In particular, Roennenberg et al. [42] suggest active therapeutic interventions to promote self-efficacy, through education, relaxation and mindfulness, self-help, and physical activation rather than passive organ-related measures. Multimodal treatment is reserved for more severe cases. They suggest that research is needed into prevention, psychophysiology, and differential treatment for patients with different symptoms.

## Figures and Tables

**Figure 1 ijerph-17-04232-f001:**
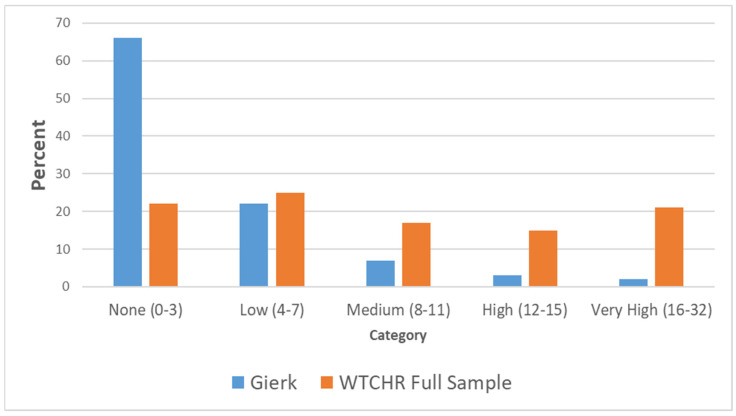
Distribution of Somatic Symptoms by Category (Note: Use color for print).

**Table 1 ijerph-17-04232-t001:** Frequency Distribution of Sociodemographic Variables and SSS-8 Mean/SD.

Characteristic	Total		SSS-8 Total Sore	
	*n*	%	Mean	SD
Total Sample	2322	100	9.72	7.17
Injury Severity				
None	1547	65.62	7.76	6.02
Low	100	4.31	11.06	6.91
Medium	369	15.89	13.74	7.26
High	306	13.18	14.29	7.54
K6 Trajectory				
Resilient	556	23.94	5.00	4.88
Low-stable	1339	57.67	9.75	6.31
Moderate increasing	226	9.73	16.20	7.16
High Decreasing	201	8.66	15.17	7.08
Gender				
Male	1315	56.63	8.99	6.86
Female	1007	43.37	10.60	7.22
Age at 9/11				
0–17	7	0.30	7.33	6.12
18–24	105	4.52	8.26	6.42
25–44	1194	51.42	10.06	7.37
45–64	998	42.98	9.39	6.76
65+	18	0.78	10.41	4.87
Race				
White	1779	76.61	9.13	6.79
Black	215	9.26	11.32	7.76
Hispanic	224	9.65	13.00	7.37
Asian	86	3.70	9.20	7.09
Other	18	0.78	7.11	6.88
Education				
Less Than or High School	330	14.21	12.37	7.60
Some College	470	20.24	11.64	7.42
College or Post-Grad	1522	65.55	8.52	6.53
Income				
0 ≤ 25 K	135	5.81	12.08	8.45
25 ≤ 50 K	334	14.38	12.14	7.70
50 ≤ 75 K	497	21.40	10.30	6.92
75 ≤ 150 K	933	40.18	9.09	6.76
≥150 K	423	18.22	7.69	6.03
Eligibility Group				
Rescue/Recovery	535	23.04	11.03	7.52
Resident	264	11.37	8.66	6.60
Area Worker	1330	57.28	9.33	6.89
Passerby	193	8.31	9.79	7.09
Marital Status at Wave 4				
Married/Living	1592	69.10	9.24	6.98
Divorced/Separated/Widowed	712	30.90	10.69	7.15
Social Support at Wave 4				
Low	1280	56.16	11.83	7.30
High	999	43.84	8.03	6.39
Stressful Events				
None	1514	65.23	8.49	6.59
Low	696	29.99	11.53	7.11
High	111	4.78	14.59	8.34
Chronic Physical Diseases–Pre/Post-9/11				
0	689	29.67	7.41	6.07
1	747	32.17	8.74	6.56
2 or more	886	38.16	12.28	7.38
Chronic Mental Diseases–Pre-9/11				
No	2131	91.77	9.61	7.11
Yes	191	8.23	10.55	6.50

**Table 2 ijerph-17-04232-t002:** Regression of Somatization on Injury Severity and K6 Trajectories.

Characteristic *	Somatic Symptoms		
	b	CI (L)	CI (U)
Injury Severity			
None	Ref		
Low	1.5	0.3	2.7
Medium	3.1	2.4	3.8
High	3.7	2.9	4.5
K6 Trajectory			
Resilient	Ref		
Low-stable	3.3	2.7	3.9
Moderate increasing	8.2	7.2	9.2
High decreasing	6.4	5.4	7.4

* Regressions controlled for age at 9/11, race, gender, education, marital status, income, eligibility group, social support, stressful events, chronic physical diseases before 9/11, and chronic mental diseases before 9/11.

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
