# Peer review of "Injury Severity and Psychological Distress Sustained in the Aftermath of the Attacks of 11 September 2001 Predict Somatic Symptoms in World Trade Center Health Registry Enrollees Sixteen Years Later"

_ijerph, 2020, doi:10.3390/ijerph17124232_

Round 1

Reviewer 1 Report

This study examined medically unexplained somatic symptoms, psychological distress, and their interplay with severity of injury among a subsample of World Trade Center Health Registry enrollees followed for approximately 15 years. The analysis sample consisted of Registry enrollees who completed all registry survey waves plus the HQoL survey, were at least 18 years old on 9/11, who spoke English, and who either incurred a specified set of injury types (cuts, abrasions, or puncture wounds; sprains or strains; fractures or dislocations; and concussions or head injuries) or who sustained no injury in the attack.  Additional inclusion criteria consisted of having been south of Chambers Street , reporting injury at 2 waves of assessment (baseline and HQoL), and having non-missing K6 values at Waves 1, 2, and 4. 

About one in five respondents had a “very high” burden of somatic symptoms as measured by the Somatic Symptom Scale-8 (SSS-8) scale. Somatic symptom burden was monotonically related both to psychological distress measured by the K6, and to the measure of injury severity. The authors conclude that survivors of 9/11 exhibit substantial burdens of somatic symptoms associated with both physical and psychological consequences of exposure to the attacks, and that physical and mental health providers need to coordinate their efforts to meet the needs of those survivors.

The enormous health burdens faced by survivors of 9/11 have been well documented. This study extends findings beyond diagnosable chronic physical (e.g., cardiovascular, respiratory) and mental health (e.g., PTSD) conditions to medically unexplained somatic symptoms. Although these are may be regarded as less “objectively serious,” they exact a heavy toll. In addition, relating medically unexplained symptoms to nonspecific psychological distress, as opposed to diagnosable conditions like PTSD, highlights the reality that psychological sequelae of trauma do not have to manifest in diagnosable disorders to impose substantial suffering.

With the above said, the study in present form contributes less than it might. The quality of writing could do with substantial improvement with regard to grammar, clarity, and conciseness. In addition, it would be appropriate to describe the methodology in greater detail, because it is not obvious that much of the journal’s readership is familiar with the studies based on the WTCHR. 

My other concerns include the following:

            1) The selection criteria for respondents in the injury survivor group seems predominantly to capture those with mild to moderate injuries. Perhaps this reflects that those with the most severe, including potentially devastating, injuries did not (or could not) enroll in the registry. Alternatively, their somatic symptoms might be more likely to be confounded with “objectively measurable” sequelae of their injuries. A clear justification for the inclusion criteria with respect to injury types, including a description of what types of casualties would have been found in the (excluded) “other injury” group, is warranted.

            2) The inclusion requirement for non-missing K6 scores at Waves 1, 2, and 4, but not 3, also warrants justification.

            3) The requirement of English language capability potentially imposes a very serious selection bias by excluding at least some members of minority race/ethnic groups substantially represented in the population living and working near Ground Zero and among whom disparities in all kinds of health-related burdens, including trauma exposures and their sequelae, have been extensively documented. Were the study to begin in present day, it is not clear that this requirement would pass muster either with funding agencies or with institutional review boards. This limitation requires treatment in the Discussion.

            4) In light of the lack of a suitable U.S.-based comparison group, I don’t understand why these investigators chose to use the SSS-8 as their measure of somatic symptom burden. Although endorsement of somatic symptoms from the NLAAS appears consistent with the SSS-8 findings from the present sample, the assessments are not comparable. Moreover, there are ethnic and cultural variations in the endorsement of somatic symptoms, including in the context of somatization of mental health conditions like anxiety and depression. Therefore, I suggest the authors take a more cautious approach in their assertions that NLAAS results mitigate the limitations of a U.S. comparison group on the SSS-8.

            5) The mental health status variable was narrowly defined to encompass self-reported diagnoses of depression, anxiety, and PTSD. It excluded externalizing conditions such as substance use disorders, ADHD, and conduct and antisocial personality disorders. These conditions, encompassing impulsivity, aggression, and rule breaking manifestations, are associated with risks of exposure to trauma. This is not in any way to engage in “victim blaming” for exposure to the terror attacks. Rather, prior trauma predicts both exposure to subsequent trauma and adverse outcomes given exposure. In addition, externalizing pathology, particularly of the antisocial variety, is associated with both diagnosed chronic disease and medically unexplained symptoms. These considerations warrant treatment in the Discussion.

            6) The definition of social support at Wave 4 as a protective factor of interest is not optimal, particularly given the definition of stressors at Wave 3 as one of the risk factors and the primary exposure occurred up to 16 years previously. Although social support is indeed well-known to buffer stress and protect against a range of adverse outcomes, poor social support at Wave 4 could be a consequence of respondents having worn down their support systems as a result of their somatic symptoms and associated impairments. If social supports were assessed at earlier waves, they should consider rerunning their analyses in a way that would take better advantage of the prospective design of the WTCHR.

            7) In reporting their analytic approach, the authors need not belabor their strategy for dealing with descriptives. On the other hand, they describe the capabilities of PROC TRAJ in general, but do not discuss the specifics of how they used the program to identify their trajectories of psychological distress. For example, they should report which mathematical forms they considered, and measures of model fit as well as subject matter criteria upon which they relied to come up with their final model.

            8) The authors report “dose-response tests.” However, they do not describe, and need to describe, exactly how they tested for dose-response relationships, and how they defined these.

            9) I am also a bit surprised that the authors did not report tests of interactions of injury severity by psychological distress. Were these performed, and, if so, with what results?

Author Response

Manuscript ijerph-799099: Injury Severity and Psychological Distress Sustained in the Aftermath of the Attacks of September 11th, 2001 Predict Somatic Symptoms in World Trade Center Health Registry Enrollees Sixteen Years Later

Reviewer’s comments

Response

Page (original IJERPH version)

Reviewer #1

1

The selection criteria for respondents in the injury survivor group seems predominantly to capture those with mild to moderate injuries. Perhaps this reflects that those with the most severe, including potentially devastating, injuries did not (or could not) enroll in the registry. Alternatively, their somatic symptoms might be more likely to be confounded with “objectively measurable” sequelae of their injuries. A clear justification for the inclusion criteria with respect to injury types, including a description of what types of casualties would have been found in the (excluded) “other injury” group, is warranted.

Although the list of injuries reported in wave 1 seem mild to moderate, in the HQoL survey injured enrollees reported the circumstances of their injury (Jacobson, 2018).  9.8% of reported being hit by a falling object, and 8.3% reported hitting their head on an object.  Further, in their answers to an open-ended question on how injury occurred enrollees provided details such as “a building fell on me” and “consumed in fire”. Since such injuries were likely to be more than moderately severe, we believe that this study includes enrollees with mild, moderate, and severe injuries.  We added appropriate material to the discussion section.

See lines 96-101

2

The inclusion requirement for non-missing K6 scores at Waves 1, 2, and 4, but not 3, also warrants justification.

 The K6 scale was administered in the wave 3 survey, but the data could not be used because unfortunately the Likert scale used was reversed in order from prior questions in the same questionnaire and resulted in largely uninterpretable responses.

See line 145

3

The requirement of English language capability potentially imposes a very serious selection bias by excluding at least some members of minority race/ethnic groups substantially represented in the population living and working near Ground Zero and among whom disparities in all kinds of health-related burdens, including trauma exposures and their sequelae, have been extensively documented. Were the study to begin in present day, it is not clear that this requirement would pass muster either with funding agencies or with institutional review boards. This limitation requires treatment in the Discussion.

The Registry assessed language in the wave 1 survey.  Fully 95.2% listed English as their spoken language.  So there were too few non-English speakers for analytic purposes, but we recognize this as a limitation for generalizing to the entire population at risk.  We have added this into the limitations section.

See lines 95-96

4

In light of the lack of a suitable U.S.-based comparison group, I don’t understand why these investigators chose to use the SSS-8 as their measure of somatic symptom burden. Although endorsement of somatic symptoms from the NLAAS appears consistent with the SSS-8 findings from the present sample, the assessments are not comparable. Moreover, there are ethnic and cultural variations in the endorsement of somatic symptoms, including in the context of somatization of mental health conditions like anxiety and depression. Therefore, I suggest the authors take a more cautious approach in their assertions that NLAAS results mitigate the limitations of a U.S. comparison group on the SSS-8.

We chose the SSS-8 as our measure of somatic symptoms because it is a well validated scale (see Gierk et al, 2014) that is brief, and therefore easy to complete (i.e. some somatic symptom scales consist of as many as 78 questions.  See the two new references in the associated text). 

We agree with the reviewer’s call for caution in making comparisons between studies using different somatic symptom scales in different countries.  We have appropriately modified the relevant section of the discussion

See lines 110-120

5

The mental health status variable was narrowly defined to encompass self-reported diagnoses of depression, anxiety, and PTSD. It excluded externalizing conditions such as substance use disorders, ADHD, and conduct and antisocial personality disorders. These conditions, encompassing impulsivity, aggression, and rule breaking manifestations, are associated with risks of exposure to trauma. This is not in any way to engage in “victim blaming” for exposure to the terror attacks. Rather, prior trauma predicts both exposure to subsequent trauma and adverse outcomes given exposure. In addition, externalizing pathology, particularly of the antisocial variety, is associated with both diagnosed chronic disease and medically unexplained symptoms. These considerations warrant treatment in the Discussion.

The reviewer’s point is well taken.  We have modified the discussion section to address the reviewer’s concerns.

See lines 150-152

6

The definition of social support at Wave 4 as a protective factor of interest is not optimal, particularly given the definition of stressors at Wave 3 as one of the risk factors and the primary exposure occurred up to 16 years previously. Although social support is indeed well-known to buffer stress and protect against a range of adverse outcomes, poor social support at Wave 4 could be a consequence of respondents having worn down their support systems as a result of their somatic symptoms and associated impairments. If social supports were assessed at earlier waves, they should consider rerunning their analyses in a way that would take better advantage of the prospective design of the WTCHR.

For our analysis we preferred measures of social support and stressors at Wave 4, closer to the time of the measurement of somatic symptoms.  That is why we used social support at Wave 4.  However, as we only measured stressor at Wave 3, we were limited for that covariate.

For the reviewer’s interest, in the analytic sample we employed, the distribution of social support was the same at waves 3 and 4.  Further, bivariate regressions of the SSS-8 scale separately on social support at waves 3 and 4 showed almost identical regression coefficients and standard errors.  So, we do not believe that employing social support from an earlier wave will substantially affect the results.

See lines 165-172

7

In reporting their analytic approach, the authors need not belabor their strategy for dealing with descriptives. On the other hand, they describe the capabilities of PROC TRAJ in general, but do not discuss the specifics of how they used the program to identify their trajectories of psychological distress. For example, they should report which mathematical forms they considered, and measures of model fit as well as subject matter criteria upon which they relied to come up with their final model.

We have added a fuller description of the use of PROC TRAJ in the methods section.

See lines 186-192

8

The authors report “dose-response tests.” However, they do not describe, and need to describe, exactly how they tested for dose-response relationships, and how they defined these.

We have added a description of how the dose-response tests were conducted into the methods section.

See lines 193-197

9

I am also a bit surprised that the authors did not report tests of interactions of injury severity by psychological distress. Were these performed, and, if so, with what results?

We regret the omission.  We performed a test of the K6 trajectory – injury severity interaction, and it was not significant (p ≈ 0.39).  We have added appropriate descriptions of the approach in the methods section and added the above results into the results section.

See lines 226-231

Reviewer 2 Report

I think this is an important and well-written article. I have only some minor issues that it would be interesting to see in the article:

What are the possible implications and further possible research questions based on this research? how can these results inform further research on the relationship between physical and psychological symptoms in other relevant contexts?

Author Response

Manuscript ijerph-799099: Injury Severity and Psychological Distress Sustained in the Aftermath of the Attacks of September 11th, 2001 Predict Somatic Symptoms in World Trade Center Health Registry Enrollees Sixteen Years Later

Reviewer’s comments

Response

Page (original IJERPH version)

Reviewer #2

What are the possible implications and further possible research questions based on this research? How can these results inform further research on the relationship between physical and psychological symptoms in other relevant contexts?

We cite Roennenberg C, et al. Functional Somatic Symptoms – Clinical Practice Guideline. Dtsch Arztebl Int 2019;116: 553-60.  They suggest active therapeutic interventions to promote self-efficacy, through education, relaxation and mindfulness, self-help, and physical activation rather than passive organ-related measures.  Multimodal treatment is reserved for more severe cases.  They suggest that research is needed into prevention, psychophysiology, differential treatment of patients with different symptoms.

Round 2

Reviewer 1 Report

No further comments.